# Effect of the Combined Compound Probiotics with Glycyrrhinic Acid on Alleviating Cytotoxicity of IPEC-J2 Cells Induced by Multi-Mycotoxins

**DOI:** 10.3390/toxins14100670

**Published:** 2022-09-27

**Authors:** Lijun Wang, Xiaomin Wang, Juan Chang, Ping Wang, Chaoqi Liu, Lin Yuan, Qingqiang Yin, Qun Zhu, Fushan Lu

**Affiliations:** 1College of Animal Science and Technology, Henan Agricultural University, Zhengzhou 450046, China; 2Institute of Animal Husbandry and Veterinary Medicine, Henan Academy of Agricultural Sciences, Zhengzhou 450003, China; 3Henan Delin Biological Product Co., Ltd., Xinxiang 453000, China; 4Henan Puai Feed Co., Ltd., Zhoukou 466000, China

**Keywords:** compound probiotics, glycyrrhinic acid, cytotoxicity alleviation, aflatoxins B1, deoxynivalenol, zearalenone

## Abstract

Aflatoxins B1 (AFB1), deoxynivalenol (DON) and zearalenone (ZEA) are the three most prevalent mycotoxins, whose contamination of food and feed is a severe worldwide problem. In order to alleviate the toxic effects of multi-mycotoxins (AFB1 + DON + ZEA, ADZ) on inflammation and apoptosis in swine jejunal epithelial cells (IPEC-J2), three species of probiotics (*Bacillus subtilis*, *Saccharomyces cerevisiae* and *Pseudomonas lactis* at 1 × 10^5^ CFU/mL, respectively) were mixed together to make compound probiotics (CP), which were further combined with 400 μg/mL of glycyrrhinic acid (GA) to make bioactive materials (CGA). The experiment was divided into four groups, i.e., the control, ADZ, CGA and ADZ + CGA groups. The results showed that ADZ decreased cell viability and induced cytotoxicity, while CGA addition could alleviate ADZ-induced cytotoxicity. Moreover, the mRNA expressions of IL-8, TNF-α, NF-Κb, Bcl-2, Caspase-3, ZO-1, Occludin, Claudin-1 and ASCT2 genes, and protein expressions of TNF-α and Claudin-1 were significantly upregulated in ADZ group; while the mRNA abundances of IL-8, TNF-α, NF-Κb, Caspase-3, ASCT2 genes, and protein expressions of TNF-α and Claudin-1 were significantly downregulated in the ADZ + CGA group. In addition, the protein expressions of COX-2, ZO-1, and ASCT2 were significantly downregulated in the ADZ group, compared with the control group; whereas CGA co-incubation with ADZ could increase these protein expressions to recover to normal levels. This study indicated that CGA could alleviate cytotoxicity, apoptosis and inflammation in ADZ-induced IPEC-J2 cells and protect intestinal cell integrity from ADZ damages.

## 1. Introduction

Mycotoxins are secondary metabolites of certain fungal species, mainly in the genera of *Aspergillus* and *Fusarium.* Mycotoxins are often coexisted and widely distributed, and possess superimposition toxicity [1]. The survey involving 24,455 samples collected from 17 Asian countries during 2008 to 2019 showed that 92% of the samples were contaminated with at least one mycotoxin, and 88% of them were co-contaminated [2]. Although more than 400 mycotoxins have been discovered to date, only several mycotoxins, including aflatoxin B1 (AFB1), deoxynivalenol (DON) and zearalenone (ZEA), have been studied intensively because of their significant impacts on agriculture, economics, and public health [3]. Among the domestic animals, pig is considered as one of the most sensitive ones to the important mycotoxins. When mycotoxins are introduced through food /feed, they first interact with the gastrointestinal tract [4,5]. The epithelium layer at the innermost of mucosa is of vital importance for intestinal barrier function [6]. It acts as a barrier to block the entry of harmful agents such as pathogens, toxins, and foreign antigens. Besides, the epithelium layer is also an important site for nutrient absorption including electrolytes, dietary nutrients, and water via its selective permeable membrane [7].

Mycotoxins have been found to alter the normal intestinal functions such as barrier function and nutrient absorption in the gastrointestinal tract. Moreover, some mycotoxins can affect the histomorphology of intestine. Like most of the mycotoxins, AFB1 can compromise the health of gastrointestinal tracts. In an *in vitro* experiment using colon cell line (Caco-2) to determine the AFB1 toxicity in the intestine, AFB1 significantly inhibited cell growth, increased lactate dehydrogenase activity and caused intestinal immune damage [8]. In an animal trial with rats, the intestinal injuries induced by AFB1 were demonstrated, including the disruption of intestinal barrier, cell proliferation, cell apoptosis, and immune system [9]. DON is mainly absorbed in the intestine [10], a decreased absorption of glucose was observed following DON intoxication resulted from suppressed SGLT1 (glucose transporter) mRNA expression [11]. Apart from the glucose absorption, SGLT1 is also responsible for water reabsorption, thus reduction of SGLT1 transporter induces diarrhea as well [12]. Toxicological studies of ZEA revealed its effects on the reproductive system. Besides, the adverse effects of ZEA on gastrointestinal tracts have been reported. ZEA plays a negative role in gut health. Swine ingested ZEA did not show changes in the height of villi, the thickness of the mucosa, and number of goblet cells [13,14,15]. The high frequency of co-occurrence of ZEA with AFB1 and DON indicates great potential of a wide range of synergistic and additive interactions among the three mycotoxins. Therefore, it is necessary to develop strategy that will be capable of e effectively mitigating co-occurrence of multiple mycotoxins, such as integrated microbial degradations.

Compare with traditional physical and chemical mycotoxin degradation methods, biodegradation has the characteristics of green and healthy, and has a great application prospect. *Bacillus subtilis* was found to be able to degrade AFB1, ZEA and DON [16,17,18]. *Saccharomyces cerevisiae* was reported to have AFB1 and ZEA degrading ability [19,20] as well as alleviate cytotoxicity of porcine jejunal epithelia cells induced by DON [21]. *Pseudomonas lactis* presented effectiveness in degrading AFB1 [22].

Glycyrrhizic acid (GA) is a major bioactive ingredient extracted from Chinese herb licorice root. It has evident pharmacological properties including detoxification, immune regulation, anti-inflammatory, antitumor and antibacterial capacity [23,24]. Furthermore, the anti-inflammatory mechanism of GA for intestinal epithelial cells is closely related to inhibition of inflammatory factor release and intervention of inflammatory signaling pathway [25,26]. The previous work from our laboratory showed that GA could alleviate DON-induced oxidative stress, inflammation, and apoptosis in IPEC-J2 cells [27].

Based on the characteristics of GA and compound probiotics (CP) including *Bacillus subtilis*, *Saccharomyces cerevisiae* and *Pseudomonas lactis*, it is speculated that the combination of CP and GA (CGA) may have synergistic functions to degrade AFB1, ZEA and DON as well as alleviate mycotoxin cytotoxicity more efficiently. In order to verify CGA functions, this research aim to explore how CGA synchronously relief of cytotoxicity apoptosis, inflammation, and impaired barrier function and nutrient transporter for IPEC-J2 cells induced by AFB1, ZEA and DON, so as to find an effective method to alleviate multi-mycotoxin hazards.

## 2. Results

### 2.1. Screening the Optimal Microbial Combinations for High Degradation Rate of Multi-Mycotoxins

The best combination of *Bacillus subtilis*, *Saccharomyces cerevisiae* and *Pseudomonas lactis* was optimized by orthogonal test design with an L9(3^3^) matrix to obtain the maximum degradation rate of mycotoxins. The experimental conditions and results are summarized in Table 1. A further orthogonal analysis was conducted, and the K and R values were calculated and listed in Table 1. According to the R values in Table 1, the degradation rate of mycotoxins depended on factor A (*Bacillus subtilis*) followed by factor B (*Saccharomyces cerevisiae*) and C (*Pseudomonas lactis*). According to the K values, K1 is the maximum value under the three factors of A, B and C, indicating that the optimum addition level of *Bacillus subtilis*, *Saccharomyces cerevisiae*, and *Pseudomonas lactis* were all 1 × 10^5^ CFU/mL (i.e., the optimal combination of *Bacillus subtilis*, *Saccharomyces cerevisiae* and *Pseudomonas lactis* was A1B1C1). The total multi-mycotoxin degradation rates reached the maximum. Based on this result, three species of microbes were mixed together at 1 × 10^5^ CFU/mL, respectively, to make compound probiotics (CP) to effectively degrade ADZ. Table 2 showed that the experimental results were valid (*p <* 0.01), and *Bacillus subtilis* had the greatest contribution to the simultaneous degradation of three kinds of mycotoxins (*p <* 0.01).

### 2.2. IPEC-J2 Cell Viability Affected by GA

Cell viability was significantly reduced by mycotoxin stimulation for 24 h, while GA could reduce the cell damage of mycotoxins. Figure 1 showed that 200–600 μg/mL GA additions significantly increased cell viability (*p* < 0.05); while ADZ addition significantly decreased cell viability (*p* < 0.05), but GA additions significantly alleviated cell damage induced by ADZ. Because there was an insignificant difference between 400 and 600 μg/mL GA additions for alleviating cell damage induced by ADZ, the optimal level of GA addition was confirmed as 400 μg/mL.

### 2.3. Effect of CP, GA or CGA on Alleviating the Viability of IPEC-J2 Cells Induced by ADZ

Figure 2 indicated that the relative cell viability was significantly decreased by ADZ addition after 24 h incubation (*p* < 0.05); however, it was increased by CP, GA, or CGA addition (*p* < 0.05). Cell viability was significantly higher in the ADZ + CGA group than in other groups (*p* > 0.05). This indicated that CGA is most effective in alleviating cytotoxicity induced by ADZ. Therefore, CGA was selected for the following research.

### 2.4. CGA CGA Alleviates ADZ-Induced Injury in IPEC-J2 Cells

The rates of cell apoptosis viable and necrotic are determined by Annexin V-FITC/PI staining. ADZ significantly reduced the rate of viable cells and increased the rate of necrotic cells and late apoptotic cells (*p* < 0.01), while CGA addition could alleviate cytotoxicity induced by ADZ, and restore cell viability (*p* < 0.01). Moreover, the difference in necrotic and later apoptotic between the ADZ and ADZ+CGA groups was not significant (*p* > 0.05). It was summarized that CGA addition could reduce ADZ-induced cytotoxicity to increase the rate of viable cells and decrease the rate of apoptotic cells (Figure 3A,B).

### 2.5. Effect of CGA on Alleviating IPEC-J2 Cell Inflammation Induced by ADZ

Figure 4A–C showed the relative mRNA expressions of inflammation genes in IPEC-J2 cells. Tumor necrosis factor-α (TNF-α) and nuclear factor kappa-B (NF-Κb) were significantly upregulated by ADZ treatment alone for 24 h (*p* < 0.05), compared with the control group; however, they were downregulated by CGA addition (*p* < 0.05). Figure 4D–G showed that the relative mRNA expressions of B-cell lymphoma-2 (Bcl-2) and cysteinyl aspartate specific protease 3 (caspase-3) (*p* < 0.05) were upregulated by ADZ alone treatments; however, caspase-3 was downregulated by CGA addition (*p* < 0.05). Bcl-2/Bax (Bcl2-associated X protein) value was downregulated by ADZ alone addition (*p* < 0.01), whereas it was increased by CGA addition. As shown in Figure 4H, ADZ exposure significantly increased TNF-α protein expression and decreased COX-2 protein expression compared with the control group (*p* < 0.01); whereas CGA addition could decrease TNF-α protein expression and increase COX-2 protein expression to restore the control level. This result preliminarily inferred that CGA could alleviate ADZ cytotoxicity to decrease cellular inflammation.

### 2.6. CGA Increasing mRNA Abundances and Protein Expressions of Tight Junction Protein Genes in Cells with ADZ-Induced IPEC-J2 Injury

Figure 5A–C showed that the relative mRNA expression of the zonula occludens-1 (ZO-1), occludin and claudin-1 were significantly upregulated in the ADZ group (*p* < 0.05), compared with the control group. Compared to the control group, CGA alone addition in IPEC-J2 cells decreased the relative mRNA abundance of ZO-1. In addition, ADZ+CGA co-culture significantly increased ZO-1 and claudin-1 mRNA expression (*p* < 0.05), compared with the other groups. As shown in Figure 5D, ADZ significantly downregulated protein expression of ZO-1 and upregulated protein expression of claudin-1, compared with the control group. However, ADZ + CGA co-incubation significantly decreased protein expression of claudin-1, compared with the control and ADZ groups.

### 2.7. CGA Altering the Nutrient-Transporter Gene mRNA and Protein Expressions in IPEC-J2 Cells Induced by ADZ

Figure 6A–D showed that the relative mRNA abundance of ASC amino acid transporter 2 (ASCT2) in the ADZ group was significantly higher than that in the control group (*p* < 0.05) but co-adding CGA + ADZ retrieved it. ADZ exposure had an insignificant effect on the facilitated glucose transporter (GLUT2) mRNA abundance. ADZ exposure significantly decreased PepT1 mRNA abundance (*p* < 0.05); however, CGA addition could retrieve the above result. Compared with the control group, the relative mRNA abundance of sodium-dependent glucose cotransporter 1 (SGLT1) was significantly downregulated in the other three groups (*p* < 0.05). The protein expression of ASCT2 in the ADZ group was significantly lower than that in the other three groups (*p* < 0.05); there were insignificant differences in PEPT1 protein expressions among the four groups (Figure 6E). It was inferred that CGA could regulate nutrient absorption and transport to some extent.

## 3. Discussion

The intestine tract is the predominant region of digestion and absorption of nutrients and also a barrier against foreign substances and mycotoxins. Mycotoxins usually enter the body through the gastrointestinal tract and can influence intestinal barrier function and immune response [6]. Pigs have been identified as the most sensitive animal species to mycotoxin exposure [28,29]. The intestinal epithelium is the first physiological barrier to mycotoxins after intake and thus can influence the toxic effect [30]. Therefore, IPEC-J2 cells were chosen as the model to study the mechanism of ADZ-induced apoptosis and inflammation and to find effective methods for alleviating the cytotoxicity of multi-mycotoxins. The previous study showed that AFB1 and ZEA had a synergistic effect when they acted on Caco-2 cells at a concentration of 10 μM, and an antagonistic effect on Caco-2 cells when AFB1 and ZEA concentrations were between 20 and 50 μM [31]. Another study showed that the viability of IPEC-J2 cells decreased significantly with the increase in AFB1 and ZEA concentrations when IPEC-J2 cells were treated with low-dose AFB1 and ZEA alone or in combination [32]. The above literature indicates that the combined effect of mycotoxins is related to their types, doses and host cells. Based on the previous results in our laboratory, the highest toxic combination of AFB1, ZEA and DON were 10, 150 and 600 μg/L, respectively, which provided a basis for studying the cumulative cytotoxicity of multi-mycotoxins in IPEC-J2 cells.

The biological detoxification method includes microbial degradation and the enzymatic method, i.e., microorganisms or enzymes are used to destroy the toxic structures of mycotoxins and produce non-toxic or low-toxic metabolites. Studies have shown that carbohydrates or mannan on the cell wall of *Saccharomyces cerevisiae* can adsorb mycotoxins, and microbial cleavage products can degrade mycotoxins [33,34]. At the same time, *Saccharomyces cerevisiae* also has a protective effect on intestinal epithelial cells through the adsorption of intestinal pathogens, neutralization of cytotoxic factors, inhibition of oxidative stress, maintenance of intestinal integrity and flora balance, activation of the immune system and so on [35]. Glycyrrhinic acid (GA) exhibits various pharmacological activities, which mainly protect intestinal cells through anti-inflammation, anti-oxidation, immune regulation, anti-virus, and cell membrane stabilization [36]. Therefore, compound probiotics with high mycotoxin-degradation ability and GA with high cell-protection ability were selected and combined to alleviate the cell damage induced by AFB1, ZEA and DON in this study.

The present study found that the viability of IPEC-J2 cells with or without ADZ was increased by GA addition, inferring that GA could enhance cell viability and alleviate ADZ-induced cell damage. Probiotics have been used extensively as animal feed additives. The assay in the present study showed that the cell viability in ADZ + CP group was significantly increased compared to the ADZ group, which is consistent with previous reports showing that probiotics, such as *Lactobacillus plantarum*, *Bacillus pumilus*, *Bifidobacterium*, and *Candida utilis* have mycotoxin-degradation or mycotoxin-adsorption abilities [37,38,39]. It can be concluded that CP could alleviate ADZ-induced cell damage, which could be due to mycotoxin being depredated or absorbed by CP [39].

Caspases are important mediators of programmed cell death. Caspase-3 catalyzes the cleavage of many key cellular proteins and is essential for chromatin condensation and DNA degradation during apoptosis. Generally, apoptotic proteins Bax and caspase-3 play important roles in regulating apoptosis [40]. It has been shown that caspase-3 is an important indicator of apoptosis and that Bax activates caspase-3 by activating its activity [39,41,42]. The present results showed that caspase-3 gene expression was downregulated and Bcl-2/Bax was increased by co-incubation of CGA and ADZ, indicating that CGA was able to alleviate cell damages induced by ADZ. This research also showed that apoptotic gene expressions and Annexin V-FITC/PI staining results were consistent in measuring the CGA effect on alleviating ADZ-induced cell damages. Studies have shown that probiotics can prevent apoptosis by altering the apoptotic signal transduction pathway in intestinal epithelial cells and inhibiting the expression of apoptosis-related genes [43,44]. The present study demonstrates that CGA could attenuate cell apoptosis caused by ADZ maybe by inhibiting the expression of some apoptosis-related genes.

Cytokine/chemokine receptors are known to stimulate a variety of pathways, including Toll-like receptors (TLRs) and downstream signaling processes. This leads to the activation of NF-kB, which is a major TLR response. The activated NF-kB drives cell proliferation-associated gene expressions and releases cytokines to activate immune responses [45,46]. Cytokines, such as IL-8, IL-6 and TNF-α are expressed in an NF-κB-dependent manner and acted primarily as pro-inflammatory signals [47]. In this study, incubation with ADZ for 24 h could significantly increase the relative mRNA abundances of IL-8 and TNF-α as well as protein expressions of NF-κB and TNF-α in IPEC-J2 cells. This was consistent with a previous study, in which DON, NIV, ZEA, and FB1 alone or in combination upregulated the mRNA expressions of pro-inflammatory cytokines, such as IL-1α, IL-1β, IL-6, IL-8 and TNF-α in IPEC-J2 cells through PKR, p38 and NF-κB signaling pathways [48,49]. COX-2 plays a pro-inflammatory role by synthesizing PGE2, it can also play an anti-inflammatory role by synthesizing PGD2 during the resolution phase of inflammation [50,51,52,53]. In the present study, COX-2 protein expression in the ADZ group was significantly lower than in the control group, but CGA addition retrieved it. Therefore, it could be inferred that when IPEC-J2 cells were co-incubated with ADZ + CGA at the resolution phase of inflammation, CGA could attenuate the inflammation caused by ADZ by inhibiting the expressions of some inflammatory-related cytokines.

The tight junction complex is composed of both transmembrane tight junction proteins (occludin and claudin) and cytosolic tight junction proteins (zonula occludin, ZO) that link transmembrane tight junction proteins to the actin cytoskeleton [54]. Tight junction proteins are often used as indicators of intestinal tight junction barriers and permeability function. Intestinal tight junction damage is closely related to bowel toxicity and mycotoxins disrupt intestinal barrier integrity and induce intestinal toxicity [55,56].

Specifically, up-regulation of claudin-1, occludin, and ZO-1 might decrease intestinal permeability caused by tight junction disruptions in weaned piglets [57,58]. This research showed that ADZ significantly upregulated the mRNA abundances of ZO-1, Occludin and Claudin-1 as well as claudin-1 protein expression, which was consistent with the previous report [21]. However, CGA addition further upregulated the mRNA abundances of ZO-1 and Claudin-1 to decrease intestinal permeability. The low intestinal permeability caused by CGA addition may be able to prevent ADZ from entering the intestinal barrier. This research showed that ADZ addition upregulated the mRNA abundance and downregulated the protein expression of ZO-1. The previous study reported that the cellular mRNA levels were elevated to compensate for the reduction in the protein expression level of ZO-1 [59], in agreement with this study. Compared with the addition of ADZ alone, CGA addition significantly increased the mRNA abundances of ZO-1 and Claudin-1, but downregulated the protein expression of claudin-1, which was consistent with the previous reports [57,60]. The inconsistency of mRNA abundance and protein expression may be due to posttranslational alterations. Because miRNA is an important and widespread molecule that post-transcriptionally regulates gene expression, it is hypothesized that the protein expression of ZO-1 and Claudin-1 might be regulated by miRNA. Previous studies have shown that miRNA can decrease intestinal tight junction protein expression by suppressing Rho–ROCK-mediated pathway activation [61,62]. When mycotoxins stimulate intestinal epithelial cells, the cell autoimmune response is activated. The addition of probiotics and CGA stimulates tight-junction protein secretion from cells to repair intercellular permeability, and enhance transmembrane epithelial electric resistance (TEER) to reduce external stimulation for improving the cellular barrier to achieve self-protection [63].

The small intestine is viewed as an organ that mediates nutrient digestion and absorption. The previous studies indicated that mycotoxins affect intestinal nutrient absorption in humans and mice [64,65,66]. The nutrient transporters, such as SGLT1, GLUT2, ASCT2 and PepT1 play important roles in glucose and amino acids uptake. In the present study, ADZ increased the mRNA abundance of ASCT2, but reduced the mRNA abundance of PepT1 and SGLT1 and the protein expression of ASCT2, indicating that ADZ exposure might destroy the structure and barrier functions of intestinal epithelial cells to hinder the transport of peptides. However, CGA co-incubation with ADZ increased the mRNA abundance of PepT1 and the protein expression of ASCT2 and decreased the mRNA abundance of ASCT2. The research has shown that ADZ caused disturbances in intestinal nutrient absorption and intestinal transit; nevertheless, CGA could alleviate ADZ-induced cell damage by improving nutrient transport. ASCT2 is an acronym standing for alanine, serine and cysteine transporter. High ASCT2 expression in highly proliferative cells, such as inflammatory and stem cells can fulfill the augmented glutamine demand [67]. The low mRNA abundance of ASCT2 by ADZ addition indicated that ADZ caused inflammation in the intestine, which is consistent with the inflammation results. Therefore, CGA co-incubation with ADZ may be able to return normal amino acid metabolism damaged by ADZ. As the major small peptide transporter of the small intestine, PEPT1 provides additional intestinal absorption capacity for neutral amino acids in the form of di- and tripeptides [68,69]. Upregulation of PEPT1 mRNA expression by the addition of CGA suggests that the passive transport of small peptides might increase to equilibrate small peptide transport. These results indicated that ADZ addition induced intestinal inflammation and apoptosis, and caused intestinal barrier dysfunction, and then affects the uptake and transport of nutrients, while CGA co-incubation with ADZ probably prevented ADZ injury by positively regulating transporter gene mRNA expression.

## 4. Conclusions

The current study demonstrated that CP and GA showed synergistic effects. CGA could protect ADZ-induced IPEC-J2 cells against injury by reducing cell inflammation and apoptosis, enhancing the barrier function of the intestines and regulating the uptake and transport of nutrients. This study provides the basis for the roles and mechanisms of CGA as a potential alternative substance for the remission of multi-mycotoxin cytotoxicity.

## 5. Materials and Methods

### 5.1. The Combinations of AFB1, ZEA and DON

AFB1, ZEA and DON with a purity >99% were purchased from Sigma-Aldrich (St. Louis, MO, USA). AFB1, ZEA and DON were selected as the three factors of Box–Behnke design. Based on the previous results in our laboratory, the optimal combined concentrations of AFB1, DON and ZEA (AFB1 + DON + ZEA, named as ADZ) to induce IPEC-J2 cell serious damages were 10, 600 and 150 μg/L, respectively.

### 5.2. Microbes and Glycyrrhinic Acid (GA) Preparation and Combination

After considering the beneficial microbes with a high ability for degrading AFB1, ZEA and DON, three species of microorganisms, such as *Bacillus subtilis* (CGMCC 1.0504), and *Saccharomyces cerevisiae* (CGMCC 2.3866) were obtained from China General Microbiological Culture Collection Center (CGMCC, Beijing, China), and *Pseudomonas lactis* (NBRC 113561) was obtained from NITE Biological Resource Center (NBRC, Chiba, Japan). The best combination of compound probiotics (CP) was screened using a three-factor and three-level orthogonal test, in which *Bacillus subtilis*, *Saccharomyces cerevisiae* and *Pseudomonas lactis* were selected as the three factors, and the visible counts of 1 × 10^5^ colony-forming units (CFU)/mL, 1 × 10^6^ CFU/mL and 1 × 10^7^ CFU/mL were set as the three levels for three species of microbes. The total reactive volumes were 5 mL including 3 mL spare MRS medium with or without ADZ, some volume of complex probiotic bacterial solution, and normal saline was used to adjust to 5 mL. There were three replications in each group. The reactive conditions were: 37 °C and 100 r/min for 24 h. The degradation rates of AFB1, ZEA and DON were used as reference indicators to obtain the optimal combination of compound probiotics.

In order to obtain the optimal concentration of GA for alleviating ADZ-induced IPEC-J2 cell toxicity, 0, 200, 400 and 600 μg/mL GA were selected. After the optimal concentration of GA was confirmed, it was combined with CP to make CGA for alleviating ADZ-induced cytotoxicity effectively for the following cell experiments.

### 5.3. Cell Culture and Treatments

The IPEC-J2 cell line was obtained from the College of Animal Science and Technology, Jiangxi Agricultural University (Jiangxi, China). IPEC-J2 cells were cultured in high-glucose Dulbecco’s modified eagle medium (HGDMEM) supplemented with 10% PBS and 1% penicillin (10,000 U/mL)-streptomycin (10 mg/mL) in an incubator at 37 °C and 5% CO_2_. Cells were cultured for 24 h before different treatments were performed. The treatment solutions were diluted with serum- and antibiotic-free HGDMEM.

### 5.4. Cell Viability Was Measured

We cultivated IPEC-J2 cells in 96-well microtiter plates for 24 h at a density of 5104 cells per well, Following removing the culture medium and the cells were washed twice with 5 mL of PBS, and subsequently cultured with CP, GA (400 μg/mL), CGA, ADZ, CP + ADZ, GA + ADZ, and CGA + ADZ for 24 h, respectively. The control group was added with the same volume of HGDMEM as the experimental group. CP, GA, CGA and ADZ were diluted with serum- and antibiotic-free HGDMEM. After the reaction, incubation at 37 °C for 4 h followed by addition of 10 μL/well of MTT solution (5 mg/mL PBS). Thereafter, the supernatant was removed, 150 mL DMSO was added to each well and the samples were shaken for 15 min. Finally, optical density was determined with a spectrophotometric microplate reader (BioTek Elx 808) (BIO-TEK Instruments Inc., Winooski, VT, USA) using a wavelength of 490 nm and background subtraction at 630 nm.

### 5.5. Annexin V-FITC/PI Apoptosis Determination

Apoptotic IPEC-J2 cells were determined by Annexin V and PI staining using an Annexin V-FITC kit (Beijing ComWin Biotech Co., Ltd., Beijing, China), in accordance with the manufacturer’s instructions. Inoculated IPEC-J2 cells were added to 6-well-plates at a density of 3 × 105 cells/well and incubation was carried out for 24 h, then PBS was used to wash the cells twice after removing the culture medium, and subsequently cultured with control, ADZ, CGA and CGA + ADZ for 24 h. After that, Cells were digested with 0.25% trypsin and EDTA, then centrifuged for 5 min at 800 rpm. A PBS wash was performed twice to remove debris and pancreatin from the cells, cells were then resuspended in 100 μL binding buffer; then, the cells were incubated with 5 μL Annexin V-FITC and stained with 10 μL PI at room temperature for 15 min in the dark. At last, the cell pellets were resuspended in 400 μL PBS each and immediately analyzed by flow cytometry (Em 530 nm and Ex 488 nm, Becton Dickinson Company, Franklin Lakes, NJ, USA).

### 5.6. RNA Extraction and Quantitative Real-Time PCR

At a density of 3 × 10^5^ cells/well, cells were plated into 6-well culture plates and allowed to adhere for 24 h.

After four treatments (control, ADZ, CGA, ADZ + CGA) for 24 h. According to the manufacturer’s instructions, total RNA was extracted with TRIzol^®^ reagent (Invitrogen, Carlsbad, CA, USA). An RNase-free solution was blended with RNA pellets and stored at 80 °C for future analysis. Measurement of RNA concentration was carried out with a NanoDrop ND-1000 spectrophotometer at OD260/OD280 and OD260/OD230. RNA was visualized with an agarose gel to check its integrity. TB GREEN kit (TaKaRa, Dalin, China) was used to reverse transcribe 1 μg samples of total RNA into cDNA. RT-PCR was performed in triplicates in 20 μL total volume in a CFX Connect™ Real-Time System (Bio-Rad, Hercules, CA, USA). The PCR cycle conditions were 95 °C for 300 s, 95 °C for 30 s, followed by 38 cycles of 60 °C for 40 s, 72 °C for 30 s, 95 °C for 10 s, and 65 °C for 60 s. PCR products were tested for specificity and reliability using a melting curve analysis. In Table 3, you can find a list of all primers that were used in this study. To normalize the data to the control sample, glyceraldehyde-3-phosphate dehydrogenase (GAPDH) was utilized as a housekeeping gene. The 2^−ΔΔCT^ method was used to analyze the data from RT-PCR [70].

### 5.7. Western Blotting Analysis

After four kinds of treatments, each group of cells was extracted with RIPA buffer (EpiZyme Biotechnology, Shanghai, China) to obtain total proteins; by using a BCA Protein Assay Kit (Sangon Biotech, Shanghai, China), the concentrations of protein were measured. SDS-PAGE was used to separate equal amounts of total proteins, and PVDF membranes were used to transfer them. The membranes were blocked in TBST with 5% milk powder for 2 h at room temperature (RT). Next, primary antibodies were incubated overnight at 4 °C, followed by horseradish peroxidase-conjugated secondary antibodies for 2 h at RT. A method of enhanced chemiluminescence was used to visualize the protein bands, and the data were analyzed using Image Quant Software (Amersham Bioscience, Amersham, UK). Finally, the relative expression of proteins was normalized to the expression of β-actin.

### 5.8. Data Analysis and Statistics

Experimental data were represented as the mean ± standard errors. The data were analyzed using the SPSS software (SPSS 25.0, IBM, Armonk, NY, USA), and one-way ANOVA was used to compare treatment means. Duncan’s test was used to analyze multiple comparisons. Statistical significance was determined by *p <* 0.05. All graphs were generated using GraphPad Prism 8 (GraphPad Software, La Jolla, CA, USA).

## Figures and Tables

**Figure 1 toxins-14-00670-f001:**
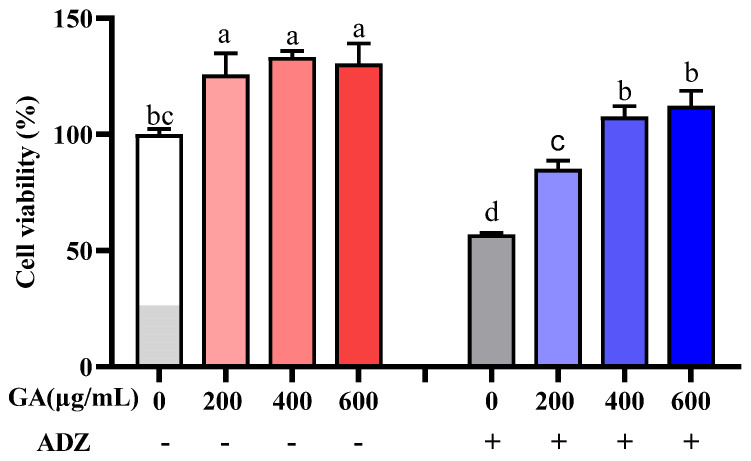
Effect of co-incubation of GA with ADZ on IPEC-J2 cell viability. White and red bars represent IPEC-J2 cells incubated with GA for 24 h, dark grey and blue bars represent IPEC-J2 cells incubated with GA and ADZ for 24 h. The different lowercase letters on each bar indicate significant differences from each other (*p <* 0.05), while the same letters on each bar indicate insignificant differences from each other (*p >* 0.05). “-” indicates without ADZ addition; “+” indicates ADZ (AFB1 10 μg/L + DON 600 μg/L + ZEA 150 μg/L) addition.

**Figure 2 toxins-14-00670-f002:**
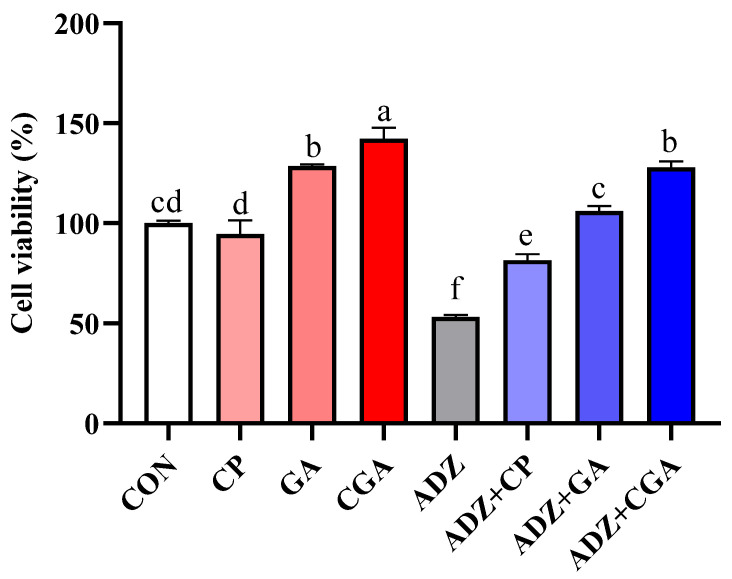
Effect of CP, GA, and CGA on IPEC-J2 cell viability decrease caused by multi-mycotoxins. White and red bars represent IPEC-J2 cells incubated with CP, GA, and CGA for 24 h; dark grey and blue bars represent IPEC-J2 cells incubated with ADZ, ADZ + CP, ADZ + GA and ADZ + CGA for 24 h. The different lowercase letters on each bar indicate significant differences from each other (*p* < 0.05), while the same letters on each bar indicate insignificant differences from each other (*p* > 0.05). CP: Compound probiotics (*Bacillus subtilis*, *Saccharomyces cerevisiae* and *Pseudomonas lactis* are 1 × 10^5^ CFU/mL, respectively); GA: Glycyrrhizic acid (400 μg/mL); ADZ: AFB1 10 μg/L + DON 600 μg/L + ZEA 150 μg/L; CGA: CP + GA.

**Figure 3 toxins-14-00670-f003:**
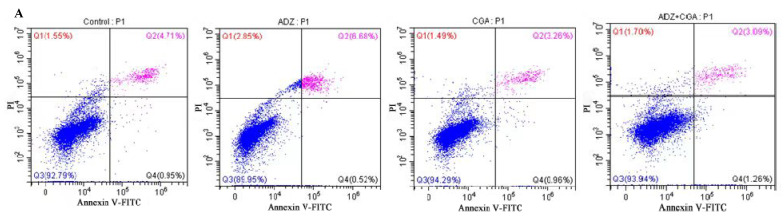
CGA remission damage of IPEC-J2 cell induced by ADZ. Note: (**A**) (c) Flow cytometry analysis of apoptotic/necrotic cells using Annexin V-FITC/PI. The red, green, blue and purple colors stand for the necrotic cell rates (Q1), the later apoptotic cell rates (Q2), the viable cell rates (Q3), and the early apoptotic cell rates (Q4), respectively. (**B**) The quantitative analysis of the necrotic, viable cell rates, early apoptosis and later apoptosis. CON: control (IPEC-J2 cells were treated with HGDMEM), ADZ: AFB1 10 μg/L + DON 600 μg/L + ZEA 150 μg/L; CGA: CP + GA. The culture time is 24 h. Data are expressed as mean ± SD with three independent experiments. The different lowercase letters on each histogram indicate significant differences from each other (*p <* 0.05), while the same lowercase letters on each histogram indicate insignificant differences from each other (*p >* 0.05).

**Figure 4 toxins-14-00670-f004:**
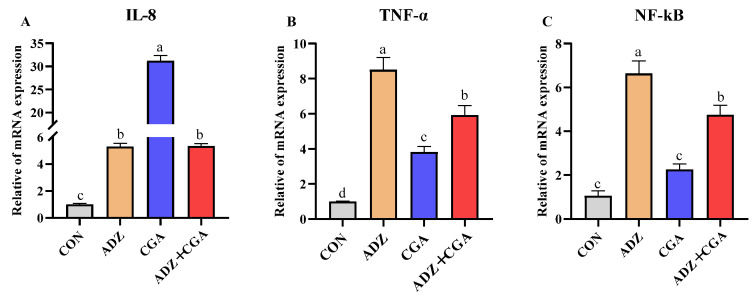
Effect of CGA on the relative mRNA and protein expressions of ADZ-induced IPEC-J2 cells inflammation and apoptotic genes. (**A**–**G**) The mRNA expression of IL-8, TNF-α, NF-κB, Bax, Bcl-2, Caspase-3 and Bax/Bcl-2. (**H**) The protein expressions of TNF-α and COX-2. CON: control (IPEC-J2 cells were cultured with HGDMEM), ADZ: AFB1 10 μg/L + DON 600 μg/L + ZEA 150 μg/L; CGA: CP + GA. The culture time is 24 h. Data are expressed as mean ± SD with three independent experiments. The different lowercase letters on each histogram indicate significant differences from each other (*p <* 0.05), while the same lowercase letters on each histogram indicate insignificant differences from each other (*p >* 0.05).

**Figure 5 toxins-14-00670-f005:**
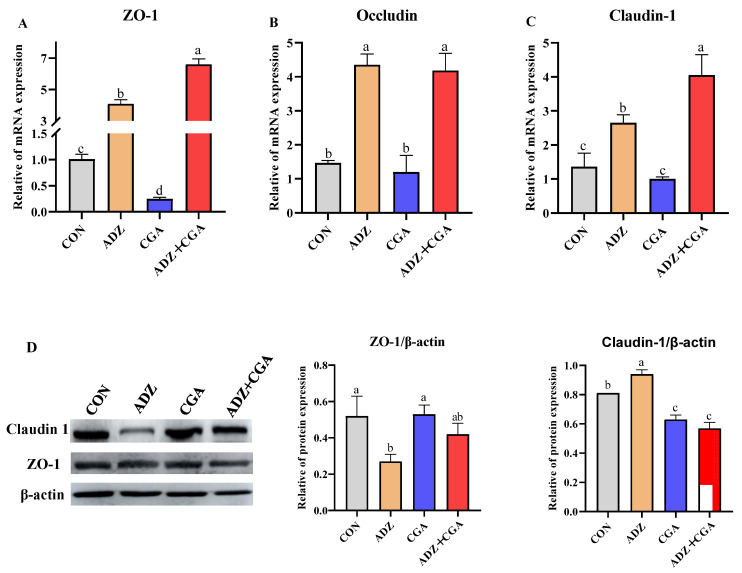
Effect of CGA on the relative of mRNA and protein expressions of tight junction protein genes in cells with ADZ-induced IPEC-J2 injury. (**A**–**C**): The mRNA expression of ZO-1, occludin and claudin-1. (**D**): The protein expressions of ZO-1 and claudin-1. CON: Control (IPEC-J2 cells were cultured with HGDMEM); ADZ: AFB1 10 μg/L + DON 600 μg/L + ZEA 150 μg/L; CGA: CP + GA. The culture time is 24 h. Data are expressed as mean ± SD with three independent experiments. The different lowercase letters on each histogram indicate significant differences from each other (*p <* 0.05), while the same lowercase letters on each histogram indicate insignificant differences from each other (*p >* 0.05).

**Figure 6 toxins-14-00670-f006:**
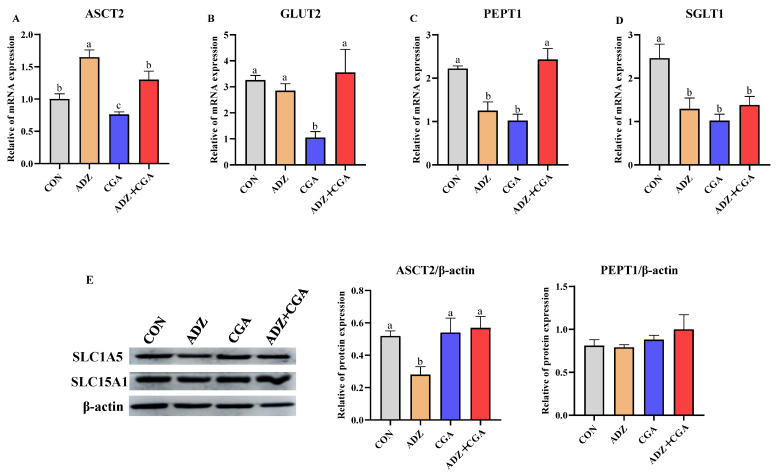
Effect of CGA on the relative of mRNA and protein expressions of nutrient-transporter genes in IPEC-J2 cells induced by ADZ. (**A**–**D**): The mRNA expressions of ASCT2, GLUT2, PepT1 and SGLT1. (**E**): The protein expressions of ASCT2 and PepT1. CON: Control (IPEC-J2 cells were cultured with HGDMEM); ADZ: AFB1 10 μg/L + DON 600 μg/L + ZEA 150 μg/L; CGA: CP + GA. The culture time is 24 h. Data are expressed as mean ± SD with three independent experiments. The different lowercase letters on each histogram indicate significant differences from each other (*p <* 0.05), while the same lowercase letters on each histogram indicate insignificant differences from each other (*p >* 0.05).

**Table 1 toxins-14-00670-t001:** Degradation rates of AFB1, ZEA and DON by different combinations of *Bacillus subtilis*, *Saccharomyces cerevisiae* and *Pseudomonas lactis* (%).

Group	Factors			Mycotoxin-Degradation Rates			
	A	B	C	AFB1	ZEA	DON	Total
1	1	1	1	43.05 ± 2.40 ^a^	38.26 ± 4.35 ^ab^	80.49 ± 0.75 ^ab^	161.80 ± 6.89 ^a^
2	1	2	2	26.85 ± 0.64 ^b^	39.21 ± 9.14 ^ab^	78.37 ± 0.40 ^ab^	144.44 ± 8.91 ^a^
3	1	3	3	28.16 ± 5.63 ^b^	46.66 ± 3.66 ^a^	82.62 ± 1.29 ^a^	157.44 ± 10.43 ^a^
4	2	1	2	42.55 ± 3.18 ^a^	28.18 ± 1.72 ^bcd^	75.69 ± 3.05 ^bc^	146.42 ± 8.99 ^a^
5	2	2	3	31.12 ± 8.92 ^b^	4.78 ± 2.21 ^e^	76.37 ± 1.26 ^bc^	112.36 ± 8.38 ^b^
6	2	3	1	8.08 ± 2.46 ^d^	37.03 ± 10.85 ^abc^	79.39 ± 0.15 ^ab^	124.50 ± 8.34 ^b^
7	3	1	3	28.34 ± 7.30 ^b^	16.91 ± 5.35 ^d^	71.52 ± 7.40 ^c^	116.77 ± 15.23 ^b^
8	3	2	1	14.82 ± 8.41 ^cd^	25.35 ± 7.23 ^cd^	77.22 ± 1.16 ^abc^	117.39 ± 15.62 ^b^
9	3	3	2	20.36 ± 5.98 ^bc^	23.08 ± 4.00 ^d^	76.97 ± 4.52 ^abc^	120.41 ± 12.56 ^b^
K1	463.68	424.99	403.69				
K2	383.29	374.19	411.27				
K3	354.57	402.35	386.58				
R	109.11	50.80	24.69				
Importance order	A > B > C						
The optimal solution	A1B1C1						

Note: The different lowercase letters in the same column indicate significant differences (*p* < 0.05), while the same lowercase letters in the same column indicate insignificant differences (*p* > 0.05). AFB1, DON and ZEA concentrations are 10, 600 and 150 μg/L, respectively. Factors A, B and C are representatives of three factors, i.e., A: *Bacillus subtilis*; B: *Saccharomyces cerevisiae*; C: *Pseudomonas lactis;* 1, 2, and 3 (under factors) are representatives of three levels of bacteria, i.e., 1 × 10^5^, 1 × 10^6^ and 1 × 10^7^ CFU/mL, respectively. Ki (i = 1, 2, 3) represents the sum of the experimental data for the impact of factor I (I =A, B, C) on the level of i (i = 1, 2, 3). The optimum addition level is determined by K value, The highest K value in the same column indicates that this level is the optimum addition level. R is the difference between the highest and lowest values of K within one factor. The importance of each factor is determined by R value, in which the higher R value in one factor indicates that this factor is more important.

**Table 2 toxins-14-00670-t002:** Main effect analyses of mycotoxin degradation by different bacterium.

Error Source	Sum of Squares	Degree of Freedom	Mean Square	F Value	*p*-Value
Corrected Model	8012.57	6	1335.43	9.32	0.00
A	6397.16	2	3198.58	22.33	0.00
B	1295.41	2	647.71	4.52	0.02
C	320.01	2	160.00	1.12	0.35
Error	2864.83	20	143.24		
Corrected Error	10,877.40	26			

Note: A, B and C are representatives of three factors, i.e., A: *Bacillus subtilis*; B: *Saccharomyces cerevisiae*; C: *Pseudomonas lactis*.

**Table 3 toxins-14-00670-t003:** Primer sequence for real time PCR.

Primer	Primer Sequence (5′-3′)	Accession Number
GAPDH	F: ATGACCACAGTCCATGCCATC	XM-004387206
	R: CCTGCTTCACCACCTTCTTG	
Bcl-2	F: AGAGCCGTTTCGTCCCTTTC	XM-003122573.2
	R: GCACGTTTCCTAGASTGCAT	
Bax	F: ATGATCGCAGCCGTGGACACG	XM-003355975.1
	R: AASTAGATGGTCACCGTCTGC	
Casepase-3	F: TTGGACTGTGGGATTGAGACG	NM-214131.1
	R: CGCTGCACAAAGTGACTGGA	
IL-8	F: GACCCCAAGGAAAAGTGGGT	NM_213867.1
	R: TGACCAGCACAGGAATGAGG	
TNF-α	F: TTCCAGCTGGCCCCTTGAGC	NM_214022
	R: GAGGGCATTGGCATACCCAC	
NF-κB	F: CTCGCACAAGGAGACATGAA	NC_006509021
	R: ACTCAGCCGGAAGGCATTAT	
ZO-1	F: CCTGAGTTTGATAGTGGCGTTGA	XM-003353439.2
	R: AAATAGATTTCCTGCCCAATTCC	
Occludin	F: ACCCAGCAAASTCATA	NM_001163647.2
	R: TCAASTTAAASTGCATA	
Claudin-1	F: ATTTCAGGTCTGGCTATCTTAGTTGC	NM_001244539.1
	R: AGGGCCTTGGTGTTGGGTAA	
ASCT2	F: CTGGTCTCCTGGATCATGTGG	DQ231578.1
	R: CAGGAAGCGGTAGGGGTTTT	
PepT1	F: CAGACTTASTCCACAACGGA	NM_214347.1
	R: TTATCCCGCCAGTACCCAGA	
SGLT1	F: TCATCATCGTCCTGGTCGTCTC	NM-34044.1
	R: CTTCTGGGGCTTCTTGAATGTC	
GLUT2	F: ATTGTCACAGGCATTCTTGTTAGTCA	NM_001097417.1
	R: TTCACTTGATGCTTCTTCCCTTTC	

## Data Availability

The data presented in this study are available in this article.

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
