# Peer review of "Effect of the Combined Compound Probiotics with Glycyrrhinic Acid on Alleviating Cytotoxicity of IPEC-J2 Cells Induced by Multi-Mycotoxins"

_toxins, 2022, doi:10.3390/toxins14100670_

Round 1

Reviewer 1 Report

Review on:

“Effect of the combined compound probiotics with glycrrhinic acid on alleviating cytotoxcicity of multi-mycotoxins”

Manuscript ID: toxins-1850076

Summary:

The authors tested the effect of glycrrhinic acid and a combination of probiotic bacteria (B. subtilis; S. cerevisiae; P. lactis) on intestinal porcine IPECJ2 cells challenged with a mix of three mycotoxins (AFB1+DON+Zea). The readout parameters after 24h treatment are:

*Cell viability (MTT)

*Nekrosis/Apoptosis (FACS)

*Inflammation related mRNA expression (IL-8;TNFalpha;NFkappaB;Bax;Bcl-2;Caspase-3)

* Inflammation related proteins (TNFalpha;Cox-2)

*Tight junction components on mRNA level (ZO-1;Occludin;Claudin-1)

*Tight junction components on protein level (ZO-1;Claudin-1)

*Transporters on mRNA level (ASCT2;Glut2;PepT1;SGLT1)

*Transporters  on protein level (SLC1A5; SLC15A1)

The authors found beneficial effects of a combined application.

Comments and questions:

Table 1: I do not really understand table 1. Does “Group 1-9“ represent different concentrations of mycotoxins or different concentrations of bacteria?

The mycotoxin-degradation rates are experimental data. How long was the incubation time? I could not find any indication in “Materials and methods” on the experimental and analytical setup leading to these data.

Figure 1/Table 3: The authors did not analysed the effect of CGA without ADZ. Why? Line lost in pdf? In the following experiments, CGA was included. A diagram similar to Fig. 1 would be helpful.

The cell viability of GA (400 µg/mL) + ADZ is approximately 50% versus control (Fig. 1). In Tab 3 the same conditions were reported with 110% viability. Could you please explain the discrepancy?

Figure 2: PE-A channel is PI? FITC-A channel is Annexin V-FITC?

Figure 5: Are the effects on ASCT2/GLUT2/PepT1/SGLT1 mRNA also found on the protein level? Were the effects on SLC1A5/A1 proteins also found on mRNA level? The authors should include these experiments if possible. At least the mRNA level of SLC1A5/A1 should be no problem.

Minor:

Figure 3 G:  Ratio of relative mRNA…….

Figure 3 H:  Ratio of relative protein…….

Figure 4 D:  Ratio of relative protein…….

Line 137: Annexin V-FITC/PI

Line 147: Annexin V-FITC/PI

Line 188: occluden-1  claudin-1 ?

Line 276: Annexin V-FITC/PI

Line 394: Cell viability was measured (MTT assay)

Line 444: “ß-actin … was used as a house-keeping gene…..” . I guess, this sentence belongs to point 5.6.

Reviewer 2 Report

The manuscript entitled “Effect of the combined compound probiotics with glycyrrhinic acid on alleviating cytotoxicity of multi-mycotoxins” have described  alleviate toxic effects of multi-mycotoxins (AFB1+DON+ZEA, ADZ) on inflammation and apoptosis of swine jejunal epithelial cells (IPEC-J2), three species of bacteria: Bacillus subtilis, Saccharomyces cerevisiae and Pseudomonas lactis. Moreover, the mRNA abundances of IL-8, TNF-α, NF-Κb, Bcl-2, Caspase-3, ZO-1, Occludin, Claudin-1 and ASCT2 genes, and protein expressions of TNF-α and Claudin-1 were estimated.

The manuscript is interesting but requires changes. Authors should correct manuscript according to the suggestion.

Minor issues:

-          In manuscript please correct microorganisms name, it should by italic. Moreover in manuscript please correct inoculum size “105, 106, 107”, I think it should be : “105, 106 and 107

-          Table 3 – “word “superscribe” should be deleted, in table are homogenic group (a, b, c) but not in supercribe

-          Materials and methods line 432: please explain how was RT-PCR results interpreted?

Reviewer 3 Report

The manuscript " Effect of the combined compound probiotics with glycyrrhinic  acid on alleviating cytotoxicity of multi-mycotoxins " presents original research results. I believe that the research was implemented correctly based on appropriate research methods. The manuscript was prepared carefully. However, I have a few comments to consider before publication. 

1.   The title of the paper should be written accurately. The research was conducted only in vitro. In my opinion, this should be stated in the title of the paper.

2.   The title of Table 1 is not very precise. It indicates that different variants of probiotics were used. It should be specified which ones? E.g. different concentrations (1×105, 1×106 and 1×107 CFU/mL).  There is a group heading in the table. What does it mean? The test groups were 1-9, while the test factor labeled A,B,C were numbered as 5,6,7. Is group 5,6,7 the same as factor 5,6 and 7? The markings are not very clear and confusing it is necessary to correct this.

3.   Table 2 Not sure what it refers to. What did Main effect analyses refer to? Title should be changed to be precise.

4.   Table 3. Effect of CP, GA or CGA on alleviating the viability of IPEC-J2 cells induced by ADZ. The title is vague abbreviations were used which should be expanded. What did the authors mean when they wrote: Effect of co-incubation of CP, GA or CGA? Interchangeably used CP and GA against CGA?

5.   The designation of the control object in the tables and charts should be unified as control.

6.   Figure 5. E. Relative protein expression of SLC15A1/β-actin - no letter designations showing statistical differences between averages.

7.   The reference in qPCR was the GAPDH gene? Please indicate this in section 5.6. Please check the given accession number XM-004387206, which refers to Florida manatee.

Minor comments:

L. 372-373. Italicize species names (in all parts of the text). Provide isolate codes and collection name.

L. 386. Following cell experiments? State which cell lines were used in the experiments. Were they intestinal porcine enterocytes cells?

Reference 1. It is misquoted: Authors' names are: Gajęcki instead of Gajcki and Gajęcka instead of Gajcka.
